# Molecular species identification of bushmeat recovered from the Serengeti ecosystem in Tanzania

Megan A. Schilling[1,2☯], Anna B. Estes[1,3☯], Ernest Eblate[3,4☯], Andimile Martin[3☯], Dennis Rentsch[5], Robab Katani[1,2,6], Asteria Joseph[7], Fatuma Kindoro[7], Beatus Lyimo[3], Jessica Radzio-Basu[6], Isabella M. Cattadori[1,3,6,8], Peter J. Hudson[1,6,8], Vivek Kapur[1,2,3,6], Joram J. Buza[3]*, Paul S. Gwakisa[3,7]*

1 The Huck Institutes of the Life Sciences, Pennsylvania State University, University Park, Pennsylvania, United States of America, 2 Department of Animal Science, Pennsylvania State University, University Park, Pennsylvania, United States of America, 3 The School of Life Sciences and Bioengineering, Nelson Mandela African Institution of Science and Technology, Arusha, Tanzania, 4 Tanzania Wildlife Research Institute, Arusha, Tanzania, 5 Lincoln Park Zoo, Conservation and Science Department, Chicago, Illinois, United States of America, 6 Applied Biological and Biosecurity Research Laboratory, Pennsylvania State University, University Park, Pennsylvania, United States of America, 7 Sokoine University of Agriculture, Morogoro, Tanzania, 8 Department of Biology, Pennsylvania State University, University Park, Pennsylvania, United States of America

☯ These authors contributed equally to this work.
* paul.gwakisa@nm-aist.ac.tz (PSG); joram.buza@nm-aist.ac.tz (JJB)

**Data Availability Statement:** The sequences of the cytochrome B gene for each sample may be found on GenBank accession numbers MN969689-MN969916.

## Abstract

Bushmeat harvesting and consumption represents a potential risk for the spillover of endemic zoonotic pathogens, yet remains a common practice in many parts of the world. Given that the harvesting and selling of bushmeat is illegal in Tanzania and other parts of Africa, the supply chain is informal and may include hunters, whole-sellers, retailers, and individual resellers who typically sell bushmeat in small pieces. These pieces are often further processed, obscuring species-identifying morphological characteristics, contributing to incomplete or mistaken knowledge of species of origin and potentially confounding assessments of pathogen spillover risk and bushmeat offtake. The current investigation sought to identify the species of origin and assess the concordance between seller-reported and laboratory-confirmed species of origin of bushmeat harvested from in and around the Serengeti National Park in Tanzania. After obtaining necessary permits, the species of origin of a total of 151 bushmeat samples purchased from known intermediaries from 2016 to 2018 were characterized by PCR and sequence analysis of the cytochrome B (*Cyt*B) gene. Based on these sequence analyses, 30%, 95% Confidence Interval (CI: 24.4–38.6) of bushmeat samples were misidentified by sellers. Misreporting amongst the top five source species (wildebeest, buffalo, impala, zebra, and giraffe) ranged from 20% (CI: 11.4–33.2) for samples reported as wildebeest to 47% (CI: 22.2–72.7) for samples reported as zebra although there was no systematic bias in reporting. Our findings suggest that while misreporting errors are unlikely to confound wildlife offtake estimates for bushmeat consumption within the Serengeti ecosystem, the role of misreporting bias on the risk of spillover events of endemic zoonotic infections from bushmeat requires further investigation.

**Funding:** This research was supported by grant from the U.S. Department of Defense, Defense Threat Reduction Agency, Biological Threat Reduction Program, Project # HDTRA1-16-1-0005 (V.K and J.J.B.).

**Competing interests:** The authors have declared that no competing interest exists.

## Introduction

Bushmeat, meat and organs from wildlife species, is an important source of animal protein in the diets of communities in the sub-tropics of the Americas and Africa. The hunting and harvesting of bushmeat is commonly practiced by those living in close proximity to national parks and protected areas and is an important source of income for rural and forest dwellers who depend on this natural resource [1]. This is particularly true for many rural populations in Tanzanian where it found that bushmeat makes its way to urban markets and neighboring countries. Non-specific snare poaching is the most common form of bushmeat hunting in the Serengeti ecosystem in Tanzania [2] and often targets migratory wildebeests, taking off an estimated 100,000–140,000 individuals per year, or 6–10% of their population [3]. Unfortunately, studies of bushmeat populations often focus on limited species. Offtake estimates are typically based on either markets, which focus on species that are for sale, or censuses counts, which typically consider only a single species. Recent modeling studies have begun to evaluate multiple species however, the interspecific dynamics of hunting remain poorly understood [4].

Globally, mammalian biodiversity is declining, with many wildlife species under threat of dramatic population losses due to the combined effects of habitat loss and illegal hunting. The effects of bushmeat hunting are estimated to put over 300 terrestrial mammal species at risk of extinction [5]. Bushmeat hunting not only threatens wildlife populations, but consumption of bushmeat also poses a significant threat to human health–of the 60% of emerging diseases that are zoonotic, more than 70% originate in wildlife [6]. This study focused on the Serengeti National Park in Northern Tanzania, which is well known for the abundance and biodiversity of wildlife within the ecosystem and the high prevalence of bushmeat hunting performed regularly in the region [7]. Further, the presence of potentially zoonotic pathogens in Tanzania has been well documented [8–13], however, such studies do not regularly include the ramifications of zoonotic transfer of pathogens or risks of disease associated with the handling and consumption of bushmeat.

Given the link between human population densities and offtake of wildlife for bushmeat, the nationwide human population increase of 2.8% per year in Tanzania and up to 3.5% per year near the Serengeti park boundary [14], is particularly concerning, and constitutes a substantial threat to the viability of wildlife populations inside and outside protected areas. Due to the illegal nature of bushmeat hunting in Tanzania, the supply chain may involve numerous players between the hunter and consumer, which could include, whole-sale sellers, retailers, and other community members who all have the potential to contribute to the mistaken or incomplete knowledge of the species of origin [1, 15–18]. Opportunities for morphological identification of bushmeat are limited since specimens are either sold as small pieces or processed in a manner that obscures indicative features. Furthermore, the reliability of the information given by the seller may also be affected by the desire to satisfy consumer preference [17, 19]. Therefore, verbal information accompanying bushmeat may not be credible. Species misinformation may have substantial effects on the response to identified spillover events or the source ascertainment of detected pathogens of human concern. Further, misreported bushmeat species may poses a threat to conservation efforts and species offtake estimates.

Nucleotide sequences in genomic regions, such as those of the mitochondrial cytochrome (*Cyt*B) gene, have been successfully used to identify a diverse set of species in studies ranging from epidemiological surveys to forensic screening of legal and illegal trade in wildlife and wildlife products [20–23]. Hence, the molecular identification of species presents an opportunity to test the reliability of seller-reported species of origin.

Due to the limitations involved with seller-reported species of origin of bushmeat and the need for correct species identification for disease surveillance and offtake estimates, this study

applied PCR amplification and DNA sequencing of the *Cyt*B gene to test the accuracy of seller-reported versus laboratory-confirmed species of origin of bushmeat collected from in and around the Serengeti ecosystem in Tanzania.

## Methods

### Study area

The Serengeti Ecosystem (Serengeti) extends from northern Tanzania into southern Kenya (1˚ 30´ to 3˚30´ S and 34˚ 00´ to 35˚ 45´ E), and covers some 25,000 km$^2$. The ecosystem is well known for its diversity of wildlife and the annual migration of ~1.5 million wildebeest (*Connochaetes taurinus mearnsi)* and ~200,000 zebra (*Equus quagga*). The Serengeti National Park in Tanzania and the Masai Mara National Reserve in Kenya have the highest levels of protection, excluding all human uses except for tourism and research, and form the core of the protected area system. In Tanzania, the Serengeti National Park is bordered by a protected area known as the Ngorongoro Conservation Area. It is also bordered by game reserves, including the Maswa Game Reserves, Kijereshi Game Reserves, and Ikorongo-Grumeti Game Reserves, which allow trophy hunting, game-controlled areas, and wildlife management areas, along with certain other human uses including livestock grazing, farming, and habitation. A distinct rainfall gradient from 500 mm/year in the southeast to 1200 mm/year in the northwest helped shape the livelihoods of people living around the ecosystem, with trans-human pastoralism dominant in the east, and agro-pastoralism dominant in the west[14].

### Bushmeat sample collection

Permissions and permits to collect samples in the Serengeti National Park was granted through the Tanzania Commission for Science and Technology, Tanzania Wildlife Research Institute (permit number TWRI/RS-331/VOL.II/2013/58 and TWRI/RS-331/VOL.II/2013/88), Tanzania National Parks (permit number TNP/HQ/C.10/1), Ngorongoro Conservation Area Authority (permit number NCAA/D/240/Vol.XXVIII/54), and Tanzania Wildlife Authority (permit number CB.517/519/01/14). Before sample collection took place, enumerator networks were formed through obtaining proper permissions from the district and village governments as well. Given the illegal nature of the bushmeat trade, local enumerators are better able to acquire the samples without raising the suspicions of those hunting or selling the bushmeat. Recruitment of enumerators was based on being resident in the study villages and having prior experience in buying bushmeat. Enumerators were instructed not to purchase multiple samples from the same seller to reduce the chances of samples being from the same animal. In an effort to mitigate bias caused by the enumerator purchasing process, they were instructed to not actively source specific species, question the seller's identification of the meat or divulge the purpose for obtaining the meat. Sampling locations were recorded using hand-held Global Positioning System (GPS) devices (Fig 1).

The enumerators purchased ~250–500 g bushmeat from local sellers. Meat was categorized as either fresh or processed at the time of purchase. Samples were considered to be 'fresh' if they did not appear to be treated in any way upon visual inspection at the time of collection. Samples were considered 'processed' if they appeared to be boiled, semi-boiled, highly salted, dried, or some combination of methods when purchased. The samples were placed in double zip lock freezer bags containing non-toxic silica gel desiccant moisture absorbers/dehumidifiers and stored in -20˚C vehicle freezers prior to transferring to -20˚C solar freezers. Samples were then transported in a -20˚C vehicle freezer to the Nelson Mandela African Institution of Science and Technology (NM-AIST) in Arusha, Tanzania where they were stored at -80˚C until processing. Cold chain was ensured through temperature trackers stored with the

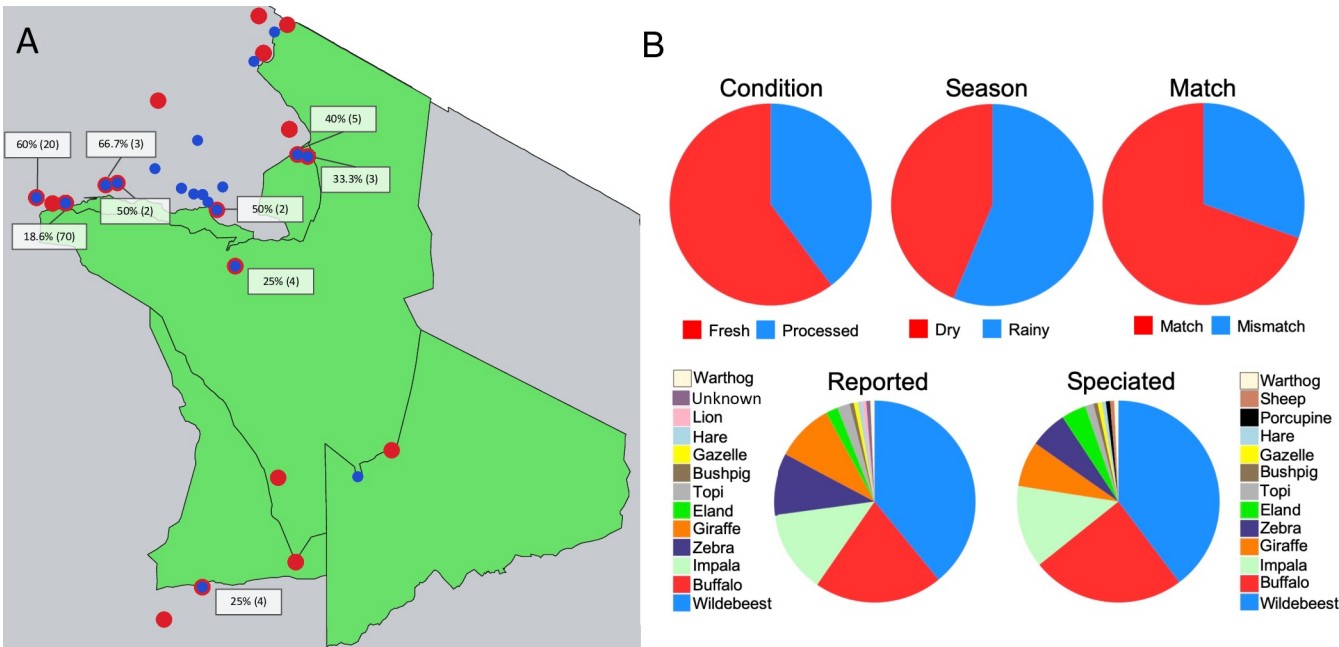

**Fig 1. Sample distribution and metadata from selected samples collected in the Serengeti ecosystem. A.** The map shows the sites from where the 151 samples selected for molecular speciation were collected. The red dots represent the samples that speciated as the same species as reported (match), and the blue dots represent samples that did not match the reported species (mismatch). The number collected at each site is represented on the map with the percentage of mismatch samples. **B.** The proportion of samples collected from each different category is represented in the pie charts. The condition was either fresh (red, n = 91) or processed (blue, n = 60), the season was dry (red, n = 66) or rainy (blue, n = 85), whether the samples matched (red, n = 105) or mismatched (blue, n = 46) the reported species. The two pie charts on the bottom represent the reported species versus the species after speciation with the species represented by the legends associated with each pie.

samples throughout the entire process. Sample identification sheets were completed for all collected samples and the metadata, including sample condition (fresh or processed), collection site (GPS units), seasonality (rainy or dry), and reported wildlife species of origin were recorded.

## DNA extraction

By sterile, disposable Rapid Core punch (World Precision Instruments, Sarasota, FL) or sterile disposable safety scalpels (VWR, Bridgeport, NJ), three small pieces (~120 mg total) from the tissue samples were dissected and placed in the MagMAX™ Lysis/Binding Solution Concentrate buffer (Thermo Fisher Scientific, Waltham, Massachusetts). Homogenization was performed using the Bead Ruptor 24 Bead Mill Homogenizer (Omni International, Kennesaw, GA) according to company's recommendations, with slight modifications. In brief, fresh samples were homogenized for 45 seconds at 5.5 m/s using 2.3 MM zirconia beads (BioSpec Products, Bartlesville, OK) in the MagMAX™ Lysis/Binding buffer. Processed samples were presoaked in the UltraPure DNase/RNase-Free Distilled Water (Thermo Fisher Scientific, Grand Island, NY) at 4°C overnight, and were homogenized for three 30-second intervals at 5.5 m/s with 2.3 MM zirconia beads in the MagMAX™ Lysis/Binding buffer. Nucleic acid extractions were performed using the KingFisher Flex automated DNA purification system (Thermo Fisher Scientific, Grand Island, NY) from tissue samples per manufacturer's instructions. If the DNA concentration was less than 4 ng/µl and/or if the quality of the amplicons analyzed by the Bioanalyzer did not pass the quality control, DNA was extracted manually using DNeasy PowerSoil Kit (Qiagen, Hilden, Germany) per manufacturer's instructions. Extracted

DNA was quantified using Qubit™ 3.0 Fluorometer (Thermo Fisher scientific, Grand Island, NY). DNA was also visualized through agarose gel electrophoresis.

## Molecular species identification

A highly informative fragment of the *Cyt*B gene was characterized by DNA sequence analysis to determine the species of origin of the collected samples. The primer pairs used in the PCR and sequencing reactions included: Forward Mcb 5′ TACCATGAGGACAAATATCATTCTG 3′ and Reverse Mcb 5′ CCTCCTAGTTTGTTAGGGATTGATCG 3′ [24]. The PCR reaction was performed in a 30 ul volume comprising of 15 ul master One *Taq* 2X Master Mix with standard buffer (New England BioLabs Inc., Ipswich, MA, USA), 20 ng genomic DNA, 0.5 μM each primer, and 8 ul nuclease free water. The cycling conditions were: initial denaturation at 95˚C for 10 min; subsequent 35 cycles of denaturation at 95˚C for 45s; annealing at 51˚C for 1 min; extension at 72˚C for 2 min and a final extension of 72˚C for 10 min [25].

The PCR products were analyzed using a 1.5% agarose gel electrophoresis to confirm presence and quality of a 480–490 bp fragment. The amplicons were purified and sequenced at the Inqaba Biotechnical Industries (Pty) Ltd, South Africa. Sequencing was carried out in both forward and reverse directions.

## Data analysis

The sequences were examined using CLC Main Work Bench v.7 (www.qiagenbioinformatics. com) to detect base calling conflicts. The forward and reverse sequences were aligned to generate consensus sequences, then compared to available sequences in the NCBI database by using BLAST tools. A cutoff point of 95% of sequence similarity [26–28] was used to identify species for each bushmeat sample, however, the top hit (in most cases > 98% similarity) was recorded as the species of origin.

All statistical analyses were performed in R v3.4.4 [29]. The kappa2 function of the IRR package was used to determine concordance between seller-reported and laboratory-confirmed results[30]. A Chi-squared test was performed to test whether the match/mismatch percentages were similar between different reported species using the prop.test function in R. The prop.test function was also used to perform pairwise analysis of the individual species (seller-reported versus laboratory-confirmed) [16]. One-sample t-test power analysis was performed to calculate the sample size with the following assumptions: power = 0.95, Significance = 0.05, an Effect size of 0.25.

## Results

A total of 151 samples were selected for molecular characterization of the wildlife species of origin. Based on the power analysis performed, a sample size of 208 was calculated however, our strategy for this study was to perform informative research and we collected opportunistic samples from a network of enumerators and from the market. We are in the process of large-scale work in Tanzania and will consider this suggestion as well as other valuable suggestions in designing a better and more informative study.

The map in Fig 1A shows the sample collection sites. Fig 1B shows the distribution of samples with respect to the sample condition, season at collection and molecular speciation (Fig 1B). Overall, roughly one third of the samples were misreported, meaning the seller-reported species of origin was not consistent with the laboratory-confirmed species of origin.

Of the 151 samples categorized using the *Cyt*B gene, 46 (30%, CI: 23.4–38.6) represented a mismatch between seller-reported and molecular species identification (Figs 2A and 3). Overall, 47% (CI: 22.2–72.2) of the bushmeat samples reported as zebra were misreported followed

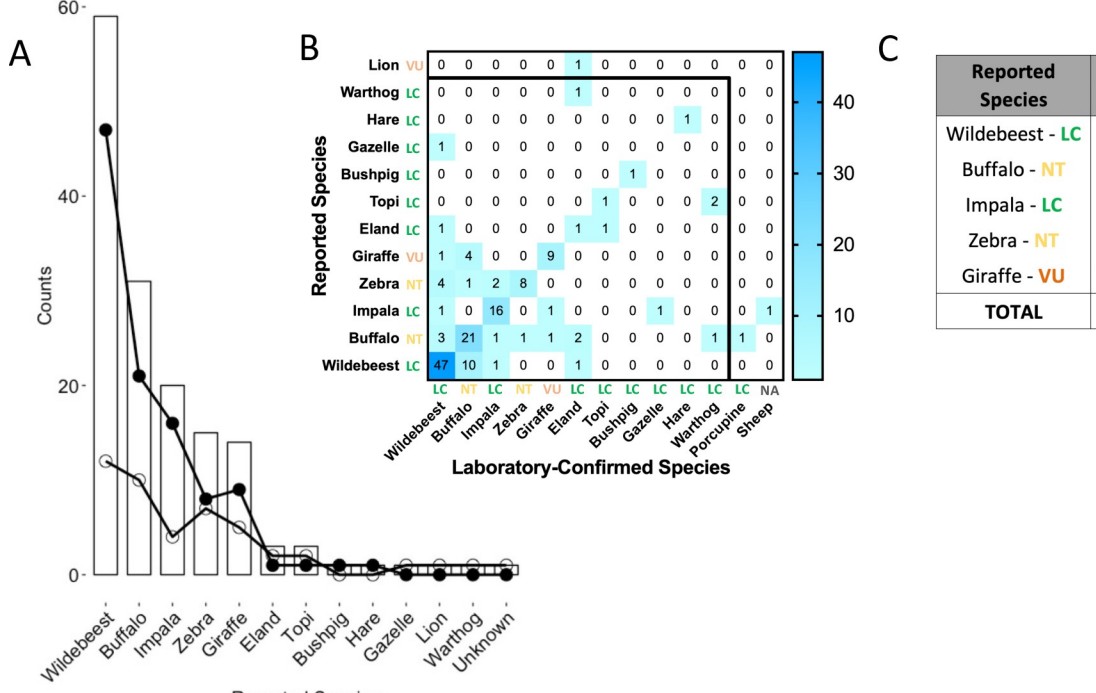

**Fig 2. A.** Bar graph and heatmap showing the number of bushmeat samples reported for each species, and the discrepancy between reported and speciated species. Total counts of seller-reported species of origin are in the bar graphs. The points and lines represent correctly reported (matched—filled circles) and mismatched (open circles) counts of the speciation results. For each species, the heatmap shows the number of seller-reported species that were confirmed with laboratory characterization, and for those misidentified, shows the actual species of each laboratory-confirmed species as well as the International Union for Conservation of Nature (IUCN) Red List Categories for each species (N = 150). The x-axis represents the laboratory-confirmed species and the y-axis represents seller-reported species. Shown IUCN categories are least concern–LC; near threatened–NT; vulnerable–VU; not applicable—NA. **B.** The percent misreporting for the top five most abundant species. The table shows the top five most abundant species and the misreporting percentage with the 95% confidence interval (CI). The graph shows the percentage as the point and the 95% CI as the range. All confidence intervals overlap, so there is no difference in misreporting percentage among the different species demonstrating no systematic bias.

by giraffe (*Giraffa camelopardalis tippelskirchii*), buffalo (*Syncerus caffer*), wildebeest, and impala (*Aepyceros melampus*) (36% CI: 14–64.4, 32% CI: 17.3–51.5, 20% CI: 11.4–33.2, 20% CI: 6.61–44.3 respectively, Fig 2A). However, there is no significant difference in the percentage of misreporting between the different species as well as compared to the overall misreporting according to a pairwise comparison of proportions (all p-values > 0.01, S1 and S2 Tables), suggesting a lack of systematic bias. For example, of the 59 wildebeest samples that were selected, 47 samples were correctly reported and 12 samples incorrectly reported as wildebeest according to the laboratory-confirmed species (Fig 2C). Of these 12 samples, 10 were laboratory-identified as buffalo, 1 as eland, and 1 as impala (Fig 2C). Overall, these data showed moderate concordance (Cohen's kappa = 0.598) between seller-reported and laboratory-confirmed species of origin. Further, there appears to be no bias with regard to the IUCN Red List designation.

Similar to the rate of mismatch of the most abundant species (Fig 2C), the overall misreporting of species from collected bushmeat samples was 30%. Despite this observation, there is no difference in the species proportions after molecular characterization (all p-values > 0.01, Fig 3) suggesting a lack of systematic bias or misreporting error (Fig 3). Fig 3A shows the number of each species as they were reported from the enumerators, for example 59 wildebeest, and the total proportion of each species of the total samples that were selected for molecular

**A**

| Species | Seller Reported | | Laboratory Confirmed | |
|---|---|---|---|---|
| | N | Percentage | N | Percentage |
| **Wildebeest** | 59 | 39.1 (31.3-47.4) | 58 | 38.4 (30.7-46.7) |
| **Buffalo** | 31 | 20.5 (14.6-28.0) | 37 | 25.5 (18.0-32.3) |
| **Impala** | 20 | 13.2 (8.47-20.0) | 20 | 13.2 (8.47-20.0) |
| **Zebra** | 15 | 9.9 (5.86-16.1) | 9 | 5.96 (2.94-11.4) |
| **Giraffe** | 14 | 9.3 (5.35-15.4) | 11 | 7.28 (3.88-13.0) |
| **Topi** | 3 | 2.0 (0.515-6.15) | 2 | 1.32 (0.23-5.20) |
| **Eland** | 3 | 2.0 (0.515-6.15) | 6 | 3.97 (1.63-8.83) |
| **Warthog** | 1 | 0.7 (0.35-4.19) | 3 | 2.0 (0.51-6.1) |

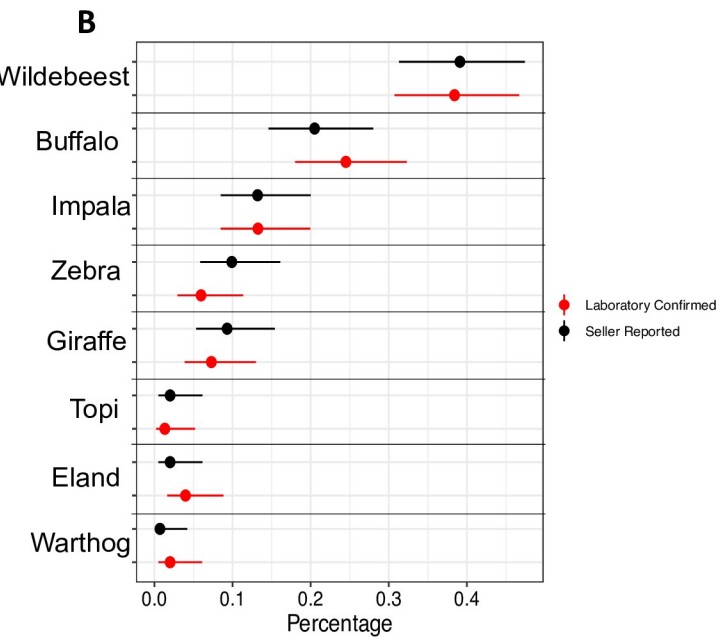

**Fig 3. Proportions of the total collected species, seller-reported species, and laboratory confirmed species.** The table shows the total number of species from which samples were collected, reported, or speciated and the proportion of the total that each species represents (Reported n = 151, Laboratory-confirmed n = 151) and the 95% confidence interval. The plot is the graphical representation of the percentage (as a proportion) of each species (the point) and the 95% CI (range). The total collected species are shown in gray, the seller-reported species in black and the laboratory-confirmed species in red.

species identification, for example 39.1% (CI: 31.3–47.4) of the samples were wildebeest (59/151). Similar numbers are shown using the results from molecular characterization (Fig 3A). For example, of the total samples tested (n = 151), 58 or 38.4% (CI: 30.7–46.7) were identified through laboratory confirmation as wildebeest suggesting no significant difference between the seller-reported or the laboratory-confirmed proportions within each species since confidence intervals are overlapping (Fig 3B).

## Discussion

Overall, approximately one third of the 151 bushmeat samples selected for laboratory-confirmation of species of origin were misidentified by the traditional methods of self-reporting. Although misreporting ranged from 20–47% (6.61–64.4) for the five most abundant species, these between species differences were not significant, suggesting that there is no predilection to misreporting one species over another. Several factors may contribute to the misreporting. Enumerators recruited and trained by project personnel recorded the species of bushmeat as reported by the seller, who may or may not have been the person who actually hunted the animal. Similarly, the seller might not actually know from which species the bushmeat was sourced or might have purposeful misreporting as a reaction to consumer preferences or differences in severity of penalties associated with the poaching of different species. For example, penalties for killing elephants (*Loxodonta africana*) are more severe than for impala [31]. Hunter preferences on where to hunt and what species to target could also influence reporting, and are in turn affected by consumer preference, threat of enforcement, severity of penalties, species abundance and proximity to the hunters' home village [22, 31]. Interestingly, misreporting is significantly more during the dry season in comparison with the rainy season (S2 Table). This difference could be explained as during the dry season, the bushmeat is typically

processed under the sun and as it gets dried, it makes it even more difficult to identify the origin.

Wildebeest was the most abundant species reported, and accounted for the most samples classified by seller-reported and molecular species identification. Since wildebeest are the most abundant ungulate in the Serengeti ecosystem, and thus heavily hunted, particularly when the migration is close to the villages surrounding the ecosystem, it is not surprising that this is the most common type of bushmeat sold in this area [3]. The misreporting of 20% (11.4–33.2) of wildebeest samples is perhaps a lower bound estimate of misreporting in this region given that it is an expected and regular source of bushmeat and may even be a preferred source of meat in this region [32]. Zebra, on the other hand, had the highest misreporting percentage (47%, 22.2–71.7), with 15 specimens reported as Zebra but only 9 confirmed as such (Fig 2). While this could simply be an artefact of the relatively small sample size as also reflected in the large confidence intervals, it may also suggest that zebra may be preferred as meat source by consumers in this region, or more susceptible to snaring than some species, or might fetch a higher price in local markets or face lower penalties or poaching than do other species, such as wildebeest (since 4 samples that speciated as wildebeest were reported as zebra; Fig 2). In the case of giraffe, the higher number of this species reported is surprising since these are the national animal symbol of Tanzania, and were seldom hunted in the ecosystem because of rumored harsh penalties [33]. However, during the past decade, there has been an increase in giraffe poaching in Serengeti, with snares set high in trees to deliberately target this large herbivorous species [33].

Unintentional misreporting of the species from the hunters to village sellers to the hands of consumer could further complicate the identification of the species. Bushmeat is presented in markets in small quantities, which make it difficult to differentiate the species from which it originated. In most cases, processed meat is more similar in appearance regardless of species, which may account for some of the misreporting if sellers are relying on morphological identification of the species [34]. Misreporting is likely a combination of lack of knowledge of the bushmeat species being sold and conscious misreporting, although the motivations for counterfeiting are not immediately clear from these data. Regardless of the reason, moderate concordance between reported species versus molecular species identification results in our study argues that validation of reported species is necessary to properly understand the dynamics of the bushmeat trade.

Accurate identification of the species being sold is particularly important for disease surveillance studies and for estimating offtake of threatened and declining species, such as the giraffe. From an ecological perspective, although the analysis was able to identify an overall relatively high (~30%) rate of misreporting of species, this is unlikely to contribute to major errors in estimations of species offtake, since the relative proportions of each species compared to the overall number of samples were not significantly different (Fig 3). However, further studies need to be conducted with larger sample sizes and specific sampling from other national parks and protected areas throughout Tanzania to confirm these findings and for more robust estimates of species offtake.

The overall ~30% misreporting rate however may potentially confound disease surveillance studies, since the species of origin in which pathogens are found plays an important role in understanding the risks of spillover and transmission dynamics associated with the pathogens. It is well documented that many zoonotic select agents, including *B. anthracis*, *Brucella spp.*, and *Coxiella* are endemic in Tanzania, and in particular in ecosystems where wildlife, livestock, and humans live in proximity increasing the chances for spillover events to occur [35–39]. It is important to note that in the Serengeti, there are many Anthrax outbreaks that affect humans, livestock, and wildlife [37]. Studies have also shown that high-risk wildlife species

that are also consumed as bushmeat including bats, rodents, and non-human primates are known to be reservoirs for viruses such as Ebola, Marburg, and Monkeypox [40–45]. Hence, misreporting bushmeat sources might considerably affect inferences and risk assessments during outbreak reporting and response [46, 47]. For instance, if a sample positive for a zoonotic pathogen is seller-reported as buffalo, but laboratory-confirmed as a zebra, this might confound both risk assessments and potential outbreak response. This is particularly relevant since the emergence and transmission of many zoonotic pathogens relies heavily on host factors and host-environment interactions, and hence the accurate identification of species in which the pathogens are present are important to understanding pathogen spillover risk and transmission dynamics [46]. In addition, since many different factors, such as environmental, ecological, and climatic differences, influence the emergence of infectious diseases and epidemiological modeling [46], accurate species identification is likely to help reduce variability caused by other confounding variables when parameterizing epidemiological models and conducting risk assessments by introducing bias, inaccurate estimates of power, and deceptive estimates of standard errors [48–50].

Limitations of this study include the small sample size for some of the minor species within the Serengeti ecosystem that precludes accurate assessments of overall misreporting risk. In addition, since the study was performed in only one ecosystem, it is unclear whether these observations can be extrapolated to other ecosystems or to the informal bushmeat markets in urban and peri-urban areas. Hence, future studies with larger sample sizes and expanding to other national parks and protected areas will provide a more robust assessment of seller-reported versus laboratory-confirmed species of origin of bushmeat in Tanzania. Finally, consideration for including consumer-preference data (which type of meat is preferred, if any), or costs of certain types of meat versus others, and any cultural or medicinal beliefs regarding the different species may help refine estimates of misreporting within the bushmeat supply chain.

## Conclusion

Although our results were from only a single ecosystem in Tanzania, they provide compelling evidence of ~30% seller-misreporting of bushmeat species. Molecular species identification of a larger number of samples in multiple ecosystems and habitat types may provide more robust and generalizable variations in misreporting and preference of species. The results also show that molecular species identification of bushmeat samples using the *Cyt*B gene sequence is a valuable tool to assist biosurveillance measures aimed at understanding the human health risks associated with the illegal harvesting and consumption of bushmeat.

## Supporting information

**S1 Table. Pairwise comparison of misreporting percentages between the top 5 most abundant species.** This table corresponds to the data and proportions in Fig 2.
(DOCX)

**S2 Table. Proportion of mismatched samples during the dry and rainy seasons.** Proportions were compared using a two-tailed Z Score analysis at a significance level of 0.05.
(DOCX)

## Acknowledgments

We thank the following agencies for permission to conduct research in the Serengeti ecosystem in Tanzania: Tanzania Commission for Science and Technology, Tanzania Wildlife Research Institute (permit number TWRI/RS-331/VOL.II/2013/58 and TWRI/RS-331/VOL.

II/2013/88), Tanzania National Parks (permit number TNP/HQ/C.10/1), Ngorongoro Conservation Area Authority (permit number NCAA/D/240/Vol.XXVIII/54), and Tanzania Wildlife Authority (permit number CB.517/519/01/14). Collection of samples would not have been possible without the cooperation of enumerators, rangers, and management authorities. Numerous laboratory technicians at the Nelson Mandela African Institution of Science and Technology and the Sokoine University of Agriculture were instrumental in the processing and analysis of samples.

## Author Contributions

**Conceptualization:** Anna B. Estes, Robab Katani, Vivek Kapur, Joram J. Buza, Paul S. Gwakisa.

**Data curation:** Megan A. Schilling, Anna B. Estes, Ernest Eblate, Andimile Martin, Robab Katani, Beatus Lyimo.

**Formal analysis:** Megan A. Schilling, Jessica Radzio-Basu, Paul S. Gwakisa.

**Funding acquisition:** Vivek Kapur, Joram J. Buza.

**Investigation:** Ernest Eblate, Andimile Martin, Asteria Joseph, Fatuma Kindoro, Beatus Lyimo.

**Methodology:** Vivek Kapur, Joram J. Buza, Paul S. Gwakisa.

**Supervision:** Anna B. Estes, Vivek Kapur, Joram J. Buza, Paul S. Gwakisa.

**Validation:** Robab Katani, Beatus Lyimo.

**Visualization:** Megan A. Schilling, Jessica Radzio-Basu.

**Writing – original draft:** Megan A. Schilling, Anna B. Estes, Robab Katani, Paul S. Gwakisa.

**Writing – review & editing:** Ernest Eblate, Andimile Martin, Dennis Rentsch, Robab Katani, Asteria Joseph, Fatuma Kindoro, Beatus Lyimo, Jessica Radzio-Basu, Isabella M. Cattadori, Peter J. Hudson, Vivek Kapur, Joram J. Buza.

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
