## [Decision Letter · Decision Letter 0]

27 Apr 2020

PONE-D-20-05756

Speciation of bushmeat recovered from the Serengeti ecosystem in Tanzania

PLOS ONE

Dear Dr Schilling,

Thank you for submitting your manuscript to PLOS ONE. After careful consideration, we feel that it has merit but does not fully meet PLOS ONE’s publication criteria as it currently stands. Therefore, we invite you to submit a revised version of the manuscript that addresses the points raised during the review process.

Both reviewers agree that your study addresses an important subject, and is timely. However, there are also significant concerns about study design and analysis that need to be addressed.

We would appreciate receiving your revised manuscript by Jun 11 2020 11:59PM. To enhance the reproducibility of your results, we recommend that if applicable you deposit your laboratory protocols in protocols.io, where a protocol can be assigned its own identifier (DOI) such that it can be cited independently in the future. For instructions see: http://journals.plos.org/plosone/s/submission-guidelines#loc-laboratory-protocols

We look forward to receiving your revised manuscript.

Kind regards,

Ulrike Gertrud Munderloh, Ph.D.

Academic Editor

PLOS ONE

Journal Requirements:

2. In your Methods section, please provide additional location information of the sampling locations, including geographic coordinates for the data set if available.

Reviewers' comments:

Reviewer's Responses to Questions

**Comments to the Author**

1. Is the manuscript technically sound, and do the data support the conclusions?

Reviewer #1: Yes

Reviewer #2: Partly

2. Has the statistical analysis been performed appropriately and rigorously? 

Reviewer #1: Yes

Reviewer #2: Yes

3. Have the authors made all data underlying the findings in their manuscript fully available?

Reviewer #1: Yes

Reviewer #2: No

4. Is the manuscript presented in an intelligible fashion and written in standard English?

Reviewer #1: Yes

Reviewer #2: Yes

5. Review Comments to the Author

Reviewer #1: This is a useful paper showing seller-perceived consumer preferences for illegal bushmeat based on misidentifications of covertly purchased bushmeat in Tanzania. I recommend this paper for publication after ra few fairly minor edits.

- Though the writing is generally of high quality, there is repeated misuse of the word 'speciation' too describe their labotratory identifications of the vertebrate species identified. Speciation refers to an evolutionary process. Therefore the authors should remove the word 'speciation' throughout the ms and replace it with molecular species identification, or something along these lines.

- In the abstract and a couple of times in the main text, common species names (e.g., wildebeest, zebra' are incorrectly capitalised.

- Line 36: Changer to "Based on these sequence analyses, 30% (96% CI:....) of bushmeat were misidentified by sellers"

- Though clear in the results section, it is not clear in the abstract (lines 38-40) whether misreporting statistics are of samples of other species misidentified as zebra or wildebeest or the reverse. Please clarify.

-Line 47: delete 'the' before 'proper'

-Line 61: Delete 'are' before 'processed'

-Lines 62-63L: "to satisfy consumer preferences"

-Line 66: Comm after 'generally'

-Line69: change to "...used to identify diverse species in studies..."

-Line 147: Replace 'using a' with 'by'

-Line 312: Delete 'the' before 'inferences'

Reviewer #2: Review of: Speciation of bushmeat recovered from the Serengeti ecosystem in Tanzania by MA Schilling et al. PLOS ONE.

This manuscript reports on a study to evaluate the levels and patterns of discordance between vendor reported and molecular genetic species identification of illegally harvested wildlife (bushmeat) in the Serengeti ecosystem of Tanzania. The motivation of the study was to evaluate the possible consequences of species misidentification for estimating harvest offtake rates and monitoring possible zoonotic disease transfer and outbreaks. While significant vendor misreporting was detected, the authors found no systematic pattern in species misidentification and concluded that harvest monitoring studies (based on market sampling) would likely be unbiased. The authors do conclude, however, that such misreporting may have serious consequences for wildlife and zoonotic disease monitoring and response. The study appears to be well conceived, but I have several concerns about the description of their basic methods, some of the analyses themselves, and the presentation of their results. I offer both major and minor comments, although several of my minor comments shouldn’t be taken lightly, in my opinion.

Major comments

Introduction. Authors do not provide a sense of the magnitude of the problem they are trying to address. Surely there are estimates of the quantity of bushmeat hunting, the degree of economic reliance or food-security contribution, the proportion of the population engaged in harvesting, transport and selling of wild game. How many villages constitute “most” (L54), what is the average number of hunters in a village? What are the market conditions in which bushmeat is sold? All or some of this information would provide the reader with a sense of the degree of risk of zoonotic disease transmission, which is stated as the primary motivation of this study. Some of this information might be better placed in the Methods-Study Area section, but providing some additional context and justification for these concerns is certainly relevant information for the introduction.

The authors do a better job making this justification in the discussion, but I think this motivation is missing in the introduction and should be included to some degree. There are many genetic papers on market species mis-identification, so a contribution related to disease transmission monitoring and response is a nice advance. From the introduction, however, this seems like a bit of a reach as there is very little offered in terms of motivation or details on the risks or history of zoonotic disease in the region. Some quantification of disease risk, rate or prevalence in wild or domestic populations, from the literature, would go a long way to further set up the rationale for this study in addition to the point above about harvest magnitude and dynamics. As it is now, I feel the authors have missed the chance to bring these 2 concepts together in the introduction to make a more impactful contribution to this literature.

L271, 298, 326, 330. The authors recognize the limitations of their inference based on small sample sizes and also mention the concern for inadequate statistical power (L323), but did not take the opportunity to conduct a power analysis themselves to estimate Type II error and determine necessary sample sizes for detecting desired effects (i.e., in risk assessment). This should be a straight-forward analysis and an important recommendation for the study to provide future researchers and resource managers.

Minor comments

L2 and elsewhere. Ok, I’ve done looked around in the literature and do see that the term ‘speciation’ or the verb ‘speciate’ have been used synchronously with ‘species identification’ but I can’t find anything authoritative (e.g., OED, Merriam Webster) on whether this is a correct usage. To me, speciation is an evolutionary process of taxon genesis rather than an observational process to distinguish among recognized species. I’ll grant that I may just be unfamiliar with the use of this term, but I’m guessing for the average ecologist, using in this way may add unnecessary ambiguity. For me the title suggested a focus on species radiation or investigation of cryptic lineages.

L27. While the manuscript title specified this work focuses on Tanzania, the abstract introduces what appear to be very general comments on bushmeat harvesting, risks of zoonotic disease, supply chains, etc. Unless the authors make these statements clearly in reference to Tanzania, they should avoid such phrases as ‘harvesting and selling of bushmeat is illegal’ because this is not true in all contexts. Other African (possibly Asian & L. American) countries allow some form of legal harvesting and selling of wild game, so caution must me taken in making such blanket statements. In L48 you add an additional caveat (proper permits) but this is not reflected in the abstract.

L63-66. Somewhat awkward phrasing with the mixing of disease detection, as the primary topic, and general conservation efforts. Suggest splitting or rephrasing.

L83. Is the fact that the Maasai don’t kill for food relevant here? Is there no risk of disease transfer (in either direction) between humans and lions (which they do hunt) that is worthy of mention?

L114, L173. What is meant by ‘processed’? The methods suggest all samples were processed in the sense of being butchered to a point where morphological features were not present. How does processed then differ from fresh and then from ‘dried’ (L124)? It wasn’t clear if by ‘dried’ the authors are referring to a function of how the collected samples were handled (i.e., silica beads) vs market condition. Please clarify your methods description to resolve these ambiguities.

L115. I understand the complications of working in African bushmeat markets, but since the primary unit of comparison with genetic species identification is that reported by the bushmeat seller, was a methodology developed, and were the sample collectors trained in how to systematically and unbiasedly elicit and record species identities in the markets? I’m not suggesting such bias exits in your data, but untrained observers can easily ask leading questions to vendors which could result in another species origin being proffered than what might have been stated. What was the protocol, for example, if one species identity was provided by a vendor but due to some morphological evidence (e.g., remaining fur, dentition, etc.) suggested an incorrect or questionable identification? Was the data collector permitted to question the identity and allow the vendor to re-identify the sample? If not permitted by the methodology, was their any attempt to evaluate whether leading questions were asked of the vendors? Reporting on how these verbal data were collected, given their basis for the subsequent analysis and conclusions, is equally if not more important than details of the cold chain process.

L157. Is a 95% similarity cutoff an appropriate measure for positive species identification? I think including some justification for this and/or a citation to other studies that have done similar. What is the distribution of sequence variability in cyt-b within and between recognized taxa? A suggestion to consider: it might be helpful to provide visual support (as well as quantified node support) for your species identification/misidentification by placing sequences in a phylogenetic tree.

L169-170. It might be helpful to provide the mean and range of samples per reported-species. Was there a minimum cutoff number of samples/reported species for species poorly represented in the market? Given that part of your analysis seems to be spatial, how many samples collected/analyzed per village location? The analyzed subset was selected based on species proportion in the collected sample, but how was the spatial distribution of these analyzed samples determined? Was it also somehow representative of the spatial distribution of the total samples collected? In L104-105, you describe recording sample collection locations but there is no description in the data analysis section of how these spatial points contribute to your analysis or findings. Generally, it is unclear the reasons for collecting the localities of sample collections and whether the spatial considerations are contributing to conclusions or inference gained.

L182 (Fig 1). There are far more blue than red dots in the figure. Are the red dots where no samples were mismatched? Just one mismatch at a location (NB: error on L183) results in a blue dot, but we don’t know what proportion of site-samples were mislabeled. More generally, do these patterns tell us anything about harvest or market dynamics?

L197-200. This is just a suggestion, but I would be curious if there is a way to test whether species reporting differed from a random assignment of species identity. Maybe a Mantel test or some sort of null-model (e.g., Ulrich & Gotelli 2012)? In addition to testing for possible bias, I think it might be interesting to evaluate how far off from just guessing meat vendors might be.

222-224. This is repetitious of what is reported earlier in results.

224-227. This is a bit confusing as written. Partly due to what appears to be a repeat of most of the table in Figure S1 and Fig 3A, with what seems to be a mistake in S1 (wildebeest, lab confirmed = 60, whereas = 58 in Fig 3A). If this is a mistake and the tables should be the same, what is the need for the repetition. If not a mistake, I am not following the difference between these tables.

L237 (Fig 3). At least for wildebeest, graph in Fig 3B appears to be backwards from the table 3A. Possibly from the confusion mentioned in previous comment?

General comment (results/discussion). Which of these species have greater national or international (e.g., CITES, IUCN) protection? I would hypothesize (a priori) that more regulated species might have been more commonly mis-identified by bushmeat sellers (i.e., possibly greater repercussions for identifying correctly). Since you do provide some hypotheses in the discussion along these lines, it might makes sense for you to include the conservation status of each species more systematically (e.g., in tabular form).

L264-265. Were there differences in wildebeest, or other species, reportings between seasons? Such information could be helpful in establishing a priori expectations for harvest composition. Since you mention this factor, you might consider reporting your observations to support this.

L272. “may be [a] preferred meat source”?

L311-312. What does “sampling must be representative of the…sample being investigated” mean?

6. PLOS authors have the option to publish the peer review history of their article (what does this mean?). If published, this will include your full peer review and any attached files.

Reviewer #1: No

Reviewer #2: No

---

## [Author Response · Author response to Decision Letter 0]

5 Jul 2020

Journal Requirements:

 Done.

2. In your Methods section, please provide additional location information of the sampling locations, including geographic coordinates for the data set if available.

Due to the sensitivity associated with bushmeat hunting in Tanzania, specific coordinates are not provided. However, the general locations of sampling are identified in the methods section and indicated on the map in Figure 1.

Done.

Reviewers' comments:

Reviewer's Responses to Questions

Comments to the Author

1. Is the manuscript technically sound, and do the data support the conclusions?

Reviewer #1: Yes

Reviewer #2: Partly

2. Has the statistical analysis been performed appropriately and rigorously?

Reviewer #1: Yes

Reviewer #2: Yes

3. Have the authors made all data underlying the findings in their manuscript fully available?

Reviewer #1: Yes

Reviewer #2: No

4. Is the manuscript presented in an intelligible fashion and written in standard English?

Reviewer #1: Yes

Reviewer #2: Yes

5. Review Comments to the Author

Reviewer #1: This is a useful paper showing seller-perceived consumer preferences for illegal bushmeat based on misidentifications of covertly purchased bushmeat in Tanzania. I recommend this paper for publication after ra few fairly minor edits.

- Though the writing is generally of high quality, there is repeated misuse of the word 'speciation' too describe their labotratory identifications of the vertebrate species identified. Speciation refers to an evolutionary process. Therefore the authors should remove the word 'speciation' throughout the ms and replace it with molecular species identification, or something along these lines.

- In the abstract and a couple of times in the main text, common species names (e.g., wildebeest, zebra' are incorrectly capitalised.

- Line 36: Changer to "Based on these sequence analyses, 30% (96% CI:....) of bushmeat were misidentified by sellers"

- Though clear in the results section, it is not clear in the abstract (lines 38-40) whether misreporting statistics are of samples of other species misidentified as zebra or wildebeest or the reverse. Please clarify.

-Line 47: delete 'the' before 'proper'

-Line 61: Delete 'are' before 'processed'

-Lines 62-63L: "to satisfy consumer preferences"

-Line 66: Comm after 'generally'

-Line69: change to "...used to identify diverse species in studies..."

-Line 147: Replace 'using a' with 'by'

-Line 312: Delete 'the' before 'inferences'

Response Reviewer 1: Thank you for your constructive comments and recommendation for publication. The authors appreciate your suggested edits and these have been included in the revised manuscript. The authors also thank you for highlighting the inappropriate use of the term speciation. This has now been changed to “molecular species identification” instead, as is now noted throughout the manuscript. 

Reviewer #2: Review of: Speciation of bushmeat recovered from the Serengeti ecosystem in Tanzania by MA Schilling et al. PLOS ONE.

This manuscript reports on a study to evaluate the levels and patterns of discordance between vendor reported and molecular genetic species identification of illegally harvested wildlife (bushmeat) in the Serengeti ecosystem of Tanzania. The motivation of the study was to evaluate the possible consequences of species misidentification for estimating harvest offtake rates and monitoring possible zoonotic disease transfer and outbreaks. While significant vendor misreporting was detected, the authors found no systematic pattern in species misidentification and concluded that harvest monitoring studies (based on market sampling) would likely be unbiased. The authors do conclude, however, that such misreporting may have serious consequences for wildlife and zoonotic disease monitoring and response. The study appears to be well conceived, but I have several concerns about the description of their basic methods, some of the analyses themselves, and the presentation of their results. I offer both major and minor comments, although several of my minor comments shouldn’t be taken lightly, in my opinion.

Major comments

Introduction. Authors do not provide a sense of the magnitude of the problem they are trying to address. Surely there are estimates of the quantity of bushmeat hunting, the degree of economic reliance or food-security contribution, the proportion of the population engaged in harvesting, transport and selling of wild game. How many villages constitute “most” (L54), what is the average number of hunters in a village? What are the market conditions in which bushmeat is sold? All or some of this information would provide the reader with a sense of the degree of risk of zoonotic disease transmission, which is stated as the primary motivation of this study. Some of this information might be better placed in the Methods-Study Area section, but providing some additional context and justification for these concerns is certainly relevant information for the introduction.

The authors do a better job making this justification in the discussion, but I think this motivation is missing in the introduction and should be included to some degree. There are many genetic papers on market species mis-identification, so a contribution related to disease transmission monitoring and response is a nice advance. From the introduction, however, this seems like a bit of a reach as there is very little offered in terms of motivation or details on the risks or history of zoonotic disease in the region. Some quantification of disease risk, rate or prevalence in wild or domestic populations, from the literature, would go a long way to further set up the rationale for this study in addition to the point above about harvest magnitude and dynamics. As it is now, I feel the authors have missed the chance to bring these 2 concepts together in the introduction to make a more impactful contribution to this literature.

Response: Thank you for the suggestion to strengthen the introduction and better setting up our study by highlighting what is known regarding bushmeat hunting in Tanzania and risks of zoonotic disease in the region. This is now done, and the scope and scale of the problem is now better addressed in the first three paragraphs of the introduction. We anticipate and plan follow-on studies to directly estimate of the prevalence of major endemic zoonotic pathogens in bushmeat recovered from this region. 

L271, 298, 326, 330. The authors recognize the limitations of their inference based on small sample sizes and also mention the concern for inadequate statistical power (L323), but did not take the opportunity to conduct a power analysis themselves to estimate Type II error and determine necessary sample sizes for detecting desired effects (i.e., in risk assessment). This should be a straight-forward analysis and an important recommendation for the study to provide future researchers and resource managers.

Response: Thank you for pointing this out. We have now done a power analysis to determine the target number of samples (see Lines 193-194 and 198-200). In brief using a significance level of 0.05, and an effect size of 0.25 we determined that a sample size of 208. We note that the current investigation represents formative research to use molecular tools to better assess species of bushmeat from opportunistic samples ascertained by a network of enumerators in the region. We agree that the results of our investigations provide and opportunity to inform future research and resource managers, and anticipate that forthcoming larger-scale, follow-on investigations of bushmeat in Tanzania in this regard. 

Minor comments

L2 and elsewhere. Ok, I’ve done looked around in the literature and do see that the term ‘speciation’ or the verb ‘speciate’ have been used synchronously with ‘species identification’ but I can’t find anything authoritative (e.g., OED, Merriam Webster) on whether this is a correct usage. To me, speciation is an evolutionary process of taxon genesis rather than an observational process to distinguish among recognized species. I’ll grant that I may just be unfamiliar with the use of this term, but I’m guessing for the average ecologist, using in this way may add unnecessary ambiguity. For me the title suggested a focus on species radiation or investigation of cryptic lineages.

Response: Good point, we agree - and this was also raised by Reviewer 1. We now use the term “molecular species identification” to more accurately represent the use of DNA sequence information identity species of origin of bushmeat samples, and this is changed throughout the manuscript.

L27. While the manuscript title specified this work focuses on Tanzania, the abstract introduces what appear to be very general comments on bushmeat harvesting, risks of zoonotic disease, supply chains, etc. Unless the authors make these statements clearly in reference to Tanzania, they should avoid such phrases as ‘harvesting and selling of bushmeat is illegal’ because this is not true in all contexts. Other African (possibly Asian & L. American) countries allow some form of legal harvesting and selling of wild game, so caution must me taken in making such blanket statements. In L48 you add an additional caveat (proper permits) but this is not reflected in the abstract.

Response: Good point. Done. We have now clarified in the abstract and throughout the revised manuscript that the harvesting and selling of bushmeat is considered to be illegal in Tanzania and the possession of bushmeat requires numerous permits from various Government ministries as were obtained (through a long process) for our studies.

L63-66. Somewhat awkward phrasing with the mixing of disease detection, as the primary topic, and general conservation efforts. Suggest splitting or rephrasing.

Response: Agreed. We have separated these concepts to clarify. (Lines 63-66)

L83. Is the fact that the Maasai don’t kill for food relevant here? Is there no risk of disease transfer (in either direction) between humans and lions (which they do hunt) that is worthy of mention?

Response: Agreed. Thank you for pointing this out and we have removed this line from the manuscript.

L114, L173. What is meant by ‘processed’? The methods suggest all samples were processed in the sense of being butchered to a point where morphological features were not present. How does processed then differ from fresh and then from ‘dried’ (L124)? It wasn’t clear if by ‘dried’ the authors are referring to a function of how the collected samples were handled (i.e., silica beads) vs market condition. Please clarify your methods description to resolve these ambiguities.

Response: To better explain these differences, we have included further details in the Methods section (Lines 131-135). In brief, samples were considered ‘fresh’ if they did not appear to be treated in any way upon visual inspection at the time of collection. Samples were considered ‘processed’ if they appeared to be boiled, semi-boiled, highly salted, dried, or some combination of methods when purchased.

L115. I understand the complications of working in African bushmeat markets, but since the primary unit of comparison with genetic species identification is that reported by the bushmeat seller, was a methodology developed, and were the sample collectors trained in how to systematically and unbiasedly elicit and record species identities in the markets? I’m not suggesting such bias exits in your data, but untrained observers can easily ask leading questions to vendors which could result in another species origin being proffered than what might have been stated. What was the protocol, for example, if one species identity was provided by a vendor but due to some morphological evidence (e.g., remaining fur, dentition, etc.) suggested an incorrect or questionable identification? Was the data collector permitted to question the identity and allow the vendor to re-identify the sample? If not permitted by the methodology, was their any attempt to evaluate whether leading questions were asked of the vendors? Reporting on how these verbal data were collected, given their basis for the subsequent analysis and conclusions, is equally if not more important than details of the cold chain process.

Response: This was also a concern of ours. To mitigate potential for unintended bias in sample ascertainment but focus on whatever was available in the “market”, enumerators were instructed not to actively source any specific species, not to question what the sellers were stating, as well as to not divulge the purpose of obtaining the bushmeat. This is now better described in the methods (Lines 124-129). 

L157. Is a 95% similarity cutoff an appropriate measure for positive species identification? I think including some justification for this and/or a citation to other studies that have done similar. What is the distribution of sequence variability in cyt-b within and between recognized taxa? A suggestion to consider: it might be helpful to provide visual support (as well as quantified node support) for your species identification/misidentification by placing sequences in a phylogenetic tree.

Response: Good point. Yes, 95% identity cut-off is indeed an appropriate and frequently used cut-off for positive species identification based on CytB gene. For instance {Schrago, Pearson, and Priorac have recently used this same cut-off, and this is now referred to in the MS (Ref #s 26-28). We did consider presenting the results as a phylogenetic tree – but given the number of taxa – the summary tables and heatmaps as presented appeared more informative and relevant. 

L169-170. It might be helpful to provide the mean and range of samples per reported-species. Was there a minimum cutoff number of samples/reported species for species poorly represented in the market? Given that part of your analysis seems to be spatial, how many samples collected/analyzed per village location? The analyzed subset was selected based on species proportion in the collected sample, but how was the spatial distribution of these analyzed samples determined? Was it also somehow representative of the spatial distribution of the total samples collected? In L104-105, you describe recording sample collection locations but there is no description in the data analysis section of how these spatial points contribute to your analysis or findings. Generally, it is unclear the reasons for collecting the localities of sample collections and whether the spatial considerations are contributing to conclusions or inference gained.

Response: Done, The mean and range is now included in Figure 3. We used an opportunistic cross-sectional sampling approach in this study with a particular focus on the Serengeti ecosystem and did not attempt to achieve a minimum number of samples per species or category as this may have biased seller and enumerator behavior. We have also updated the map in Figure 1 to reflect the proportions of mismatched samples at each site. As can be noted, visual assessment does not suggest any evidence of biased spatial distribution in match / mismatch. 

L182 (Fig 1). There are far more blue than red dots in the figure. Are the red dots where no samples were mismatched? Just one mismatch at a location (NB: error on L183) results in a blue dot, but we don’t know what proportion of site-samples were mislabeled. More generally, do these patterns tell us anything about harvest or market dynamics?

Response: Thank you for pointing this out. To better reflect the site sources for the matched vs mismatched samples, we have updated the map in Figure 1 to include proportions at each site. Additional data, well beyond the scope of the current investigations, needs to be collected as part of specifically designed future studies to derive meaningful conclusions regarding harvest or market dynamics and hence these are not presented herein.

L197-200. This is just a suggestion, but I would be curious if there is a way to test whether species reporting differed from a random assignment of species identity. Maybe a Mantel test or some sort of null-model (e.g., Ulrich & Gotelli 2012)? In addition to testing for possible bias, I think it might be interesting to evaluate how far off from just guessing meat vendors might be.

Response: Great suggestion. And indeed, we did earlier try as well and together with the statistician / ecologist on our team (IMC) revisited this matter. Unfortunately, the appropriate use of the Mantel or similar test best requires larger sample sizes in each strata to derive meaningful and robust estimates and hence was not applied in this context. 

222-224. This is repetitious of what is reported earlier in results.

Response: Agree. We have better clarified this statement to avoid repetition (Line 253-254).

“Similar to the rate of mismatch of the most abundant species (Figure 2C), the overall misreporting of species from collected bushmeat samples was 30%”.

224-227. This is a bit confusing as written. Partly due to what appears to be a repeat of most of the table in Figure S1 and Fig 3A, with what seems to be a mistake in S1 (wildebeest, lab confirmed = 60, whereas = 58 in Fig 3A). If this is a mistake and the tables should be the same, what is the need for the repetition. If not a mistake, I am not following the difference between these tables.

Response: Thank you for pointing this out. This was an error which we have corrected. To simplify the data presentation, we have added the overall percentage to Figure 3 and removed the relevant data in the supplement.

L237 (Fig 3). At least for wildebeest, graph in Fig 3B appears to be backwards from the table 3A. Possibly from the confusion mentioned in previous comment?

Response: Good catch. This is now corrected.

General comment (results/discussion). Which of these species have greater national or international (e.g., CITES, IUCN) protection? I would hypothesize (a priori) that more regulated species might have been more commonly mis-identified by bushmeat sellers (i.e., possibly greater repercussions for identifying correctly). Since you do provide some hypotheses in the discussion along these lines, it might makes sense for you to include the conservation status of each species more systematically (e.g., in tabular form).

Response: Thank you, this is a great suggestion. We have now added this information in Figure 2C as well as the Results section (Lines 234-235). 

L264-265. Were there differences in wildebeest, or other species, reporting between seasons? Such information could be helpful in establishing a priori expectations for harvest composition. Since you mention this factor, you might consider reporting your observations to support this.

Response: Thank you for this suggestion. We have now added an additional supplemental table to present this analysis.

L272. “may be [a] preferred meat source”?

Response: Done.

L311-312. What does “sampling must be representative of the…sample being investigated” mean?

Response: We have changed phrasing (Lines 339-343). 

6. PLOS authors have the option to publish the peer review history of their article (what does this mean?). If published, this will include your full peer review and any attached files.

Do you want your identity to be public for this peer review? For information about this choice, including consent withdrawal, please see our Privacy Policy.

Reviewer #1: No

Reviewer #2: No

---

## [Editor Report · Decision Letter 1]

30 Jul 2020

Molecular species identification of bushmeat recovered from the Serengeti ecosystem in Tanzania

PONE-D-20-05756R1

Dear Dr. Schilling,

We’re pleased to inform you that your manuscript has been judged scientifically suitable for publication and will be formally accepted for publication once it meets all outstanding technical requirements.

Kind regards,

Ulrike Gertrud Munderloh, Ph.D.

Academic Editor

PLOS ONE
---

## [Editor Report · Acceptance letter]

3 Sep 2020

PONE-D-20-05756R1 

Molecular species identification of bushmeat recovered from the Serengeti ecosystem in Tanzania 

Dear Dr. Schilling:

I'm pleased to inform you that your manuscript has been deemed suitable for publication in PLOS ONE. Congratulations! Your manuscript is now with our production department. 

Kind regards, 

on behalf of

Dr. Ulrike Gertrud Munderloh 

Academic Editor

PLOS ONE